# Soil Mycobiome Diversity under Different Tillage Practices in the South of West Siberia

**DOI:** 10.3390/life12081169

**Published:** 2022-07-31

**Authors:** Natalia Naumova, Pavel Barsukov, Olga Baturina, Olga Rusalimova, Marsel Kabilov

**Affiliations:** 1Institute of Soil Science and Agrochemistry, Siberian Branch of the Russian Academy of Sciences, 630090 Novosibirsk, Russia; barsukov@issa-siberia.ru (P.B.); rusalimova@issa-siberia.ru (O.R.); 2Institute of Chemical Biology and Fundamental Medicine, Siberian Branch of the Russian Academy of Sciences, 630090 Novosibirsk, Russia; baturina@niboch.nsc.ru (O.B.); kabilov@niboch.nsc.ru (M.K.)

**Keywords:** ITS region, soil fungi, Chernozem, undisturbed steppe, wheat, conventional tillage, no tillage

## Abstract

Managing soil biodiversity by reduced or no tillage is an increasingly popular approach. Soil mycobiome in Siberian agroecosystems has been scarcely studied; little is known about its changes due to tillage. We studied mycobiome in Chernozem under natural steppe vegetation and cropped for wheat by conventional or no tillage in a long-term field trial in West Siberia, Russia, by using ITS2 rDNA gene marker (Illumina MiSeq sequencing). Half of the identified OTUs were *Ascomycota* with 82% of the total number of sequence reads and showing, like other phyla (*Basidiomycota*, *Zygomycota*, *Mortierellomycota*, *Chytridiomycota*, *Glomeromycota*), field-related differential abundance. Several dominant genera (*Mortierella*, *Chaetomium*, *Clonostachys*, *Gibberella*, *Fusarium,* and *Hypocrea*) had increased abundance in both cropped soils as compared with the undisturbed one and therefore can be safely assumed to be associated with wheat residues. Fungal OTUs’ richness in cropped soils was less than in the undisturbed one; however, no tillage shifted soil mycobiome composition closer to the latter, albeit, it was still similar to the ploughed soil, despite different organic matter and wheat residue content. The study provided the first inventory of soil mycobiome under different tillage treatments in the south of West Siberia, where wheat production is an important section of the regional economy.

## 1. Introduction

Nowadays soil microbiome census conducted using state-of-the-art metagenomic techniques is an indispensable initial stage for further well-integrated and more focused ecological research and assessing soil quality [1]. Soil microbiota refers to a dynamic assemblage of microorganisms, i.e., organisms of ≤5000 μm^3^ in body volume [2], including viruses, archaea, bacteria, and eukaryotes. In terms of microbial biomass, bacteria and fungi are the main contributors, with microscopic fungi accounting for three-quarters of microbial carbon in soil [3], and harbouring vast diversity [4]. Soil fungal communities are especially important for sustaining agricultural ecosystems as they decompose plant material, cycle nutrients, etc.

Fungi in soil are known to vary at least four orders of magnitude in size, from single cells to gigantic single mycelial individuals [5]. The biodiversity concepts and indices were developed for a number of specimens, i.e., individual organisms of a certain species in a given location. Obviously, an individual organism of a soil fungus, unless it is a unicellular one, is extremely difficult, if at all possible, to determine and count. One cannot, therefore, overestimate the metagenomic approach to assessing soil mycobiome taxonomic structure and biodiversity by using molecular techniques, i.e., classifying diversity into taxa defined by nucleotide sequence similarity and then by the relative abundance of taxa-specific sequence reads in the total number of reads.

As the importance of soil biodiversity for ecosystem functioning and agricultural sustainability has been long recognised [6,7], many agricultural techniques, currently employed to sustain agricultural soils, include managing soil biodiversity by reduced, minimal, or no tillage [8]. Moreover, lower carbon losses from no till soil can also mitigate the risks associated with global warming [9]. Conversion of the undisturbed area to cropped land drastically alters the aboveground community, as well as soil physiochemical and biological environments. Consequently, such conversion also modifies the soil environment for microbial communities, shifting their biodiversity. Even virus abundance and community structure in the soil can be influenced by tillage practices and land use [10]. However, soil mycobiome in the agricultural ecosystems of Siberia so far has been scarcely studied, and nothing is known about its changes in relation to vegetation, tillage, and soil properties. The aim of the study was to reveal mycobiome composition and structure in Chernozem under natural vegetation or cropped for wheat by conventional or no tillage in the long-term field experiment in the Novosibirsk region, Russia, by using ITS region DNA diversity.

## 2. Materials and Methods

### 2.1. Experimental Site and Conditions

The field trial was conducted in the Novosibirsk region, Russia (54°4′6″ N, 79°36′3″ E) in the forest-steppe zone with a sharply continental climate. As averaged over 1991–2020 [11], the mean monthly temperature in the area of the experimental site location in October is 3.5 °C (with the average minimal and maximal temperatures of 0 and 8 °C, respectively); monthly precipitation of 31 mm with 7 days of precipitation exceeding 1 mm.

The soil is a loamy arable one classified as Luvic Endocalcic Chernozem (Aric, Siltic), while the soil of remains of undisturbed (unaffected by human activity) areas with natural vegetation is classified as Luvic Endocalcic Chernozem (Siltic), according to the World Reference Base for Soil Resources [12]; Chernozem is the most common and agriculturally valuable soil type in the region.

### 2.2. Experimental Setup

The field trial was started in 2009 on an area of 40 ha. Prior to the establishment of the trial, the field was cropped under conventional tillage for >60 years. This tillage included mouldboard ploughing in the fall, and disking in the spring. Twelve years prior to the study in 2009, the field was divided into two plots. Since then, one plot has been cultivated according to the classic notill technology; whereas the top layer (down to 15 cm) of the other plot has been subjected to annual disking, which is considered conventional tillage. Both plots with conventional (CT) and no (NT) tillage were rotationally cropped (three years for spring wheat followed by a year of spring rapeseed) and simultaneously received the same rates of herbicides and fertilizers (100 kg N, 26 kg P_2_O_5,_ and 18 kg S per hectare).

In the year when soil sampling for the study was performed, spring wheat of Novosibirskaya-31 cultivar was grown. The wheat was harvested at the beginning of September 2021. The grain yield was 4.8 t ha^−1^ on the no-till field and 4.1 t ha^−1^ on the conventionally ploughed field. An undisturbed true bunchgrass steppe ecosystem (Un), adjacent to the trial fields, was also studied as a reference to the zonal soil microbiome, its vegetation being dominated by *Stipa capillata* and *Festuca valesiaca*, as well as some *Poa* spp. and *Puccinella* sp.

### 2.3. Soil Sampling and Chemical Analyses

Soil was sampled in October 2021 from the 0–5 and 5–15 cm layers in five individual replicates from each layer. In total 30 soil samples were collected and analysed.

Soil total carbon (STC) and nitrogen (STN) contents were estimated by elemental analyzer (CHNS/O 2400 Serie II, Perkin Elmer, USA); soil organic matter content was estimated by the amount of soil mass loss on ignition at 550 °C for 12 h; the content of soil labile nutrients (NO_3_^−^, exchangeable K, readily available and available P_2_O_5_) and pH (H_2_O) were measured by standard techniques [13]. Briefly, nitrate content was determined potentiometrically in 0.1% AlKSO_4_ solution (soil:solution ratio 1:5 *w*/*v*); readily available *p* and available *p* were extracted by 0.015 M K_2_SO_4_ solution (1:5 *w*/*v*) and 0.1 M (NH_4_)_2_C_2_H_4_O(COO)_2_ solution (pH = 5.7; 1:20 *w*/*v*), respectively, and determined calorimetrically. Exchangeable K was extracted by 1 M CH_3_COONH_4_ solution (pH = 7.0; 1:10 *w*/*v*) and estimated by atomic absorption spectrometer with flame atomization (Kvant-2A, Russia). Soil pH was measured by equilibrating 10 g of field-moist soil with 25 mL of deionised water. Bulk soil density was calculated as mass/volume ratio after drying a soil core of the known volume at 105 °C for 24 h. All analyses were performed in triplicates, and the data were expressed on the oven (105 °C) dry basis. Soil properties are presented in Table 1. 

### 2.4. DNA Extraction, Amplification and Sequencing

Total DNA was extracted from 0.40 g of soil using the DNeasy PowerSoil Kit (Qiagen, Germany) as per manufacturer’s instructions. The bead-beating was performed using TissueLyser II (Qiagen, Germany) for 10 min at 30 Hz. No further purification of the DNA was needed. The quality of the DNA was assessed using agarose gel electrophoresis.

The ITS2 gene marker was amplified with the primer pairs ITS3_KYO2/ITS4, combined with Illumina adapter sequences [14]. PCR amplification was performed as described earlier [15]. A total of 200 ng PCR product from each sample was pooled together and purified through MinElute Gel Extraction Kit (Qiagen, Hilden, Germany). The obtained amplicon libraries were sequenced with 2 × 300 bp paired-ends reagents on MiSeq (Illumina, CA, USA) in SB RAS Genomics Core Facility (ICBFM SB RAS, Novosibirsk, Russia). The read data reported in this study were submitted to the NCBI Short Read Archive under bioproject accession number PRJNA845814.

### 2.5. Bioinformatic Analysis

Raw sequences were analyzed with UPARSE pipeline [16] using Usearch v.11.0.667. The UPARSE pipeline included merging of paired reads; read quality filtering (-fastq_maxee_rate 0.005); length trimming (remove less 100 nt); merging of identical reads (dereplication); discarding singleton reads; removing chimeras and operational taxonomic unit (OTU) clustering using the UPARSE-OTU algorithm. The OTU sequences were assigned a taxonomy using the SINTAX [14] and ITS UNITE USEARCH/UTAX v.8.3 [17] as a reference. Taxonomic structure of thus obtained sequences was estimated by the ratio of the number of taxon-specific sequence reads (with non-fungal removed from the data matrix) to the total number of sequence reads, i.e., by the relative abundance of taxa, expressed as percentage.

The OTUs datasets were analyzed by individual rarefaction with the help of the PAST software [18]: the number of fungal OTUs detected, reaching a plateau with increasing number of sequences, showed that the sampling effort was close to saturation for all samples, thus being enough to compare biodiversity [19].

### 2.6. Statistical Analyses

Statistical analyses (descriptive statistics, ANOVA, principal components and principal coordinates analyses) were performed by using Statistica v.13.3 a (TIBCO Software Inc., Palo Alto, CA, USA) and PAST [18] software packages. OTUs-based α-diversity indices and principal coordinates analysis were calculated using PAST. Factor effects and mean differences in post-hoc comparisons by Fisher’s LSD test were considered statistically significant at the *p* ≤ 0.05 level.

## 3. Results

### 3.1. Mycobiome Taxonomic Diversity

After quality filtering, chimera, and other domains’ sequences removal, a total of 3185 different fungal OTUs were identified at 97% sequence identity level. More than half of the OTUs (1626, or 51%) were *Ascomycota*. *Basidiomycota* featured 568 OTUs (18%). *Chytridiomycota* was represented by 200 OTUs (6%), *Glomeromycota* had 108 OTUs (*ca.* 3%), and the other 11 of the identified phyla accounted for much fewer OTUs each. Notably, rather many OTUs (525 OTUs, or 17%) remained unclassified below the *Fungi* level. Overall, the clustered OTUs belonged to 618 genera, 340 families, 151 orders, 66 classes, and 16 phyla.

In terms of the relative abundance of the sequence reads’ numbers, the most abundant phylum was *Ascomycota*, accounting for 82% (mean) of all the studied soil samples (Table 2) and differing in its relative abundance between the undisturbed and notill fields in the 0–5 m layer. Other phyla, namely moderately dominating *Basidiomycota* and *Zygomycota,* also showed field-related differential abundance. The number of OTUs, unclassified below the domain level, was rather low.

*Sordariomycetes* were the ultimate dominant among the classes, being trice as abundant in the cropped fields as compared with the field under undisturbed natural vegetation (Table 2). The representatives of the *Dothideomycetes* class were markedly more abundant and dominating in the 0–5 cm layer of the undisturbed soil. Except for *Leotiomycetes*, other dominant fungal classes revealed tillage-related differential abundance.

At the genus level, there were many dominant genera with statistically significant differences between the fields; for most genera, their relative abundance was increased in both cropped fields as compared with the undisturbed one. In addition to the 18 dominant genera, listed in Table 3, other 137 genus-level sequence read clusters were found to have the differential abundance in soil mycobiome of the studied fields.

Just seven OTUs dominated the mycobiome of the undisturbed soil, together accounting for 20% of the total number of sequence reads; the cropped soils had 19 dominant OTUs in the ploughed field and 21 OTUs in the no-till field, accounting for 36–37% of the sequence reads. Only two dominant OTUs, namely *Pseudogymnoascus roseus* and *Trichocladium asperum*, were common in the soils of all the fields with no statistically significant difference in abundance (Figure 1). Two cropped fields, as expected, had between them seven common dominants, namely two strains of *Gibberella* sp., *Chaetomium* sp., *Clonostachys rosea*, *Mortierella elongata*, *Cordyceps memorabilis* and *Fusarium merismoides*.

### 3.2. Fungal α- and β-Biodiversity

Observed and potential (as indicated by Chao-1) species richness in the 0–5 cm layer in the NT soil was 18% higher than in the ploughed soil, both soils harbouring much less fungal species as compared with the same layer of the undisturbed soil (Table 4). An inversed pattern was observed for the mycobiome of the 5–15 cm layer. Other diversity indices, presented in Table 4, in the ploughed soil showed no depth-related difference, whereas in the undisturbed and no-till soils the difference between the layers was much more pronounced.

As for the β-biodiversity, the cropped fields were closely grouped and similarly distanced from the undisturbed soil under natural vegetation (Figure 2).

### 3.3. Fungal Taxa Relationship with Soil Properties

The structure of the dominant fungal phyla abundance (Figure 3a) shows their specific preferences for some basic soil properties, with one phylum, i.e., dominating *Ascomycota*, correlating positively with soil total carbon and nitrogen. Just one of the dominant OTUs displayed some positive correlation with soil organic matter, being mostly associated with available phosphorus and potassium (Figure 3b). *Chaetomium* sp. followed soil electric conductivity and nitrates, whereas *Trochocladium asperum* responded to pH. Based on the principal components, extracted from the matrix with fungal OTUs abundance, the location of the samples (Figure 3c) shows a clear separation of the undisturbed and cropped soils (along the principal component 1) and of the layers (mostly along the principal component 2), following the pattern of the sample separation based on the Bray-Curtis distance in (Figure 2).

## 4. Discussion

Our study provided the first inventory of soil mycobiome under different tillage treatments in the south of West Siberia, where wheat production is an important section of the regional economy. The obtained results showed unequivocally that undisturbed soil under natural vegetation and wheat cropped soils differed from each other in their mycobiome composition and structure, with cropped soils being close to each other despite different tillage regimes.

### 4.1. Soil Mycobiome: General Outline

The total number of fungal OTUs determined in our study (slightly more than three thousand) was very close to the number detected in at least one other study [20], but it is noteworthy that the reported numbers of identified fungal OTUs vary widely: from several hundred [21] to 37,449 [22], p. 5. *Ascomycota* and *Basidiomycota* were also ultimately prevailing with 82%–94% of the total abundance [20]; a similar phyla percentage was reported by Sharma-Poudyal et al., 2017 [21]. However, the latter study, also with wheat, found a significantly greater proportion of *Basidiomycota* in the notill fields versus conventionally tilled ones, whereas in our study *Basidiomycota* in the notill field was two times lower (Table 2). It is noteworthy, though, that, like [21], researchers sometimes fail to report the month of soil sampling, which in the temperate regions can be the main factor shaping the structure of decomposer fungal communities [23]. Therefore, the discrepancy may be due to the differences related to sampling time, i.e., seasonal dynamics.

### 4.2. Fungal OTUs, Common for the Undisturbed and Cropped Fields

Our finding that *Pseudogymnoascus* genus (*Pseudeurotiaceae*/*Dothideomycetes*/*Ascomycota*) was among the dominants, and its *Pseudogymnoascus roseus* was one of the two dominant OTUs common for the mycobiomes of all three fields, is very interesting, as not many species of the genus have been isolated from different environments in the Northern Hemisphere. The genus can be found in different cold environments, being isolated from soil, roots, and wood samples; and *Pseudogymnoascus roseus*, known as the most widespread decomposer fungus in maritime Antarctic soils, was recently shown to be reduced by 1–2 orders of magnitude when soils were warmed to >20 °C during summer [24]. Therefore, it is likely that *p. roseus* notable presence in the studied fields might have resulted from the sampling time, i.e., the end of the growing season with negative nighttime temperatures and plenty of dead plant material. This is indirectly corroborated by the fact that in the rather genetically close soil type (Phaeozem) sampled in July in the same region under the undisturbed birch forest with understory groundcover vegetation rather similar to the undisturbed field in this study, there were practically no *Pseudogymnoascus* [25].

Another mycobiome dominant, common in all three fields, *Trichocladium asperum (Chaetomiaceae*/*Sordariales*/*Sordariomycetes*/*Ascomycota*) was shown as dominating the fungal community in the late phase during the decomposition of grain crop residues [26]. In our study the marked presence of the species in soil mycobiome late in the season most likely can have resulted from a relatively longtime span (1.5 months) of dead plant material decomposition, preceding soil sampling.

### 4.3. Fungal Genera and OTUs Increased in the Wheat-Cropped Soils as Compared with the Undisturbed Soil

Our study found several dominant fungal genus-level clusters that differed in their relative abundance in the cropped fields from those in the undisturbed field in both soil layers: namely *Mortierella, Chaetomium*, *Clonostachys*, *Gibberella, Fusarium* and *Hypocrea*, and due to this may be safely assumed to be associated with wheat residues.

Let us briefly describe these genera. The major genus dominant in our study, namely *Mortierella (Mortierellaceae/Mortierellales/Mucoromycotina_incertae_sedis/Zygomycota)* fungi, is widespread in the bulk soil, rhizosphere, and plant tissues; its characteristic features like the ability to survive under very unfavorable environmental conditions and the utilization of carbon sources like cellulose, hemicellulose, chitin make them very valuable decomposers in agricultural soils [27]. Therefore, their increased presence in the wheat-cropped fields at the end of autumn after the harvest was most likely associated with plenty of dead wheat phytomass in the soil.

The second-ranked dominant genus, with a relative abundance of an order of magnitude higher than that in the undisturbed soil, namely *Chaetomium* of *Chaetomiaceae*/*Sordariales*/*Sordariomycetes*/*Ascomycota*, is a saprotrophic fungus, normally found in soil and organic composts [28] as it degrades cellulose and other organic materials by producing lytic enzymes [29]. As the genus representatives were shown to be the main hemicellulose degraders at the cooling stage of crop residue composting [30], its increased presence, like *Mortierella*’s, in the wheat-cropped fields at the end of autumn long after the harvest was most likely determined by the presence dead phytomass in soil.

Our finding that the representatives of the *Clonostachys* genus, namely *Cl. rosea* and *Cl. phyllophila*, were also increased in both cropped fields while being almost absent in the undisturbed one, was unexpected as the fungi are known for their strong control against numerous fungal plant pathogens, nematodes and insects [31,32], being globally widespread saprotrophs with the highest prevalence in soil. However, their increased abundance in the cropped fields might indirectly suggest the presence of wheat pathogens, which were absent under the natural steppe vegetation.

*Gibberella* (of *Nectriaceae*/*Hypocreales*/*Sordariomycetes*/*Ascomycota*) representatives (anamorph *Fusarium*) are distributed worldwide in soil, aquatic and semiaquatic environments, stored grain, and natural products; the fungi are regarded as mostly pathogenic, that often can be recovered for at least two years from wheat straw [33] and wheat-cropped soil [34]. As the latter in our study contained plenty of wheat residues after harvest, our finding that the genera predominated in both cropped fields, being scarce in the soil under natural vegetation, complies with such a role.

*Hypocrea* fungi (anamorph *Trichoderma*) of *Hypocreaceae*/*Hypocreales*/*Sordariomycetes*/*Ascomycota* are found in many soil ecosystems; they also participate in straw residue decomposition in arable soils [35], can reduce the severity of plant diseases by inhibiting plant pathogens in the soil and stimulate plant growth and tolerance to abiotic stress by interacting directly with roots [36]. Therefore, the genus increased abundance in both cropped fields in our study was most likely to be (a) straw-related, and (b) apparently beneficial for wheat residue decomposition.

### 4.4. Fungal Genera and OTUs Differentially Increased in the Undisturbed Soil

The finding that *Devriesia* (of *Ascomycota*) and *Knufia* (of *Pezizomycotina*) genera were notable dominants in the upper layer of the undisturbed soil, being negligible in its lower layer and in both layers of the cropped soil, implies (a) that the fungi can get into the soil with plant litter and hence are found only in the topsoil; (b) the genera involvement with roots of certain plant species which do not penetrate deeply into the lower soil layers, and (c) the absence of such plants in the cropped fields. Taxonomic study of *Devriesia* began recently [37], and since then the number of recorded species has reached 29. In our study the *Devriesia* genus was represented by several OTUs, all of them being attributed to *Devriesia pseudoamericana*; it was first found on the surface of some fruits [38] as a part of the sooty blotch complex of apples. Therefore it can be as well an epiphyte of some aboriginal steppe plants, entering the soil late in the growing season with its litter. The same might be true for *Knufia*, as it was found as an epiphyte on tree bark or dead twigs [39,40]. With *Rhinocladiella* (of *Ascomycota*) it is most likely another story, as this dominant genus was found to have notably increased abundance in the lower layer as compared with the upper 0–5 cm: as a representative of a genus was reported as a root endophyte [41], in our study it can be an endophyte of a plant species with deeper penetrating roots, which litter might have brought the fungus into the soil of the lower layers.

The mycobiome in the undisturbed soil under natural steppe vegetation was discriminated from its cropped counterparts by the increased abundance of such minor (with relative abundance from 0.01 to 1.0%) genera as *Calycina*, *Capnodium*, *Chalara*, *Cladophialophora*, *Eupenicillium, Lachnum*, *Leohumicola*, *Ophiocordyceps*, *Pyrenochaeta*, *Stagonospora* (all belonging to the *Ascomycota* phylum), *Glomus (Glomeromycota)*, *Mastigobasidium* and *Trechispora* (both belonging to *Basidiomycota),* and several other genus-level clusters, not explicitly classified to this taxonomic level as many ITS-fragment copies were classified only to a phylum level (*Ascomycota*). It is difficult to interpret the ecological roles of these genera in the undisturbed soil, but we tend to believe their presence to be determined by plant assemblage diversity, and hence higher phytomass and root exudate chemical diversity.

### 4.5. The Fungal Genera with Differential Abundance between the Cropped Fields

It is worth emphasizing that soil sampling for this study was performed late in October, i.e., 1.5 months after wheat harvesting, in order to a) minimize the effect of plant growth on microbial communities and b) to allow a relatively long period of plant residue decomposition, with the aim to observe the tillage treatment effect.

The CT mycobiome, as compared with the NT one, had an increased abundance of *Podospora, Chaetomium, Lecythophora, Preussia genera*, and a decreased abundance of *Clonostachys, Exophiala, Gibberella, Herpotrichia, Lophiostoma, Neonectria, Rhizopus* and *Talaromyces* in the 0–5 cm layer.

*Podospora* fungi are widely spread in the environment and possess a diverse enzymic repertoire, including ligninolytic activity [42]. *Chaetomium* is described above as a cellulose and organic matter degrader [27,28]. *Lecythophora*, widely distributed in soil, wood, vegetative matter, and polluted water saprotrophic lignocellulose-inhabiting sordariomycete (*Coniochaetaceae*/*Coniochaetales*/*Sordariomycetes/Ascomycota*) and known as an active participant at the early phase of straw residue decomposition [34], in our study was apparently involved in the same process in the cropped fields. *Preussia* genus, also found to be differentially abundant in both cropped fields as compared with the undisturbed one, is known to inhabit diverse soil ecosystems, often arid [43] or extremely arid ones [44]. The studied fields, located in the south of the forest-steppe zone, are within the rather dry and cold area and thus *Preussia* presence agrees with the existing knowledge about the fungus.

*Clonostachys* genus represents globally widespread saprotrophs with the highest prevalence in soil [30,31]; as for *Exophiala*, these fungi can live saprotrophically in multiple habitats, such as bulk soil, biological crusts, rock surfaces, air, natural water masses, and rhizosphere [45,46], and recently were reported to increase its relative abundance in soil due to the wheat straw addition [47]. *Herpotrichia* have been well studied as plant (pine) pathogens [48], and the genus increased abundance under NT necessitates deeper investigation. The representatives of the *Lophiostoma* genus, besides being spread in marine environments, are also found in epi- and/or endophytic fungal assemblages [49] and on dead stems and twigs [50]. Among *Neonectria* there are tree pathogens [51], but they cause European or apple canker by their cellulose-degrading ability, and therefore their higher abundance in the NT field is not surprising. As for *Rhizopus*, the genus was represented by just one OTU in this study, *Rhizopus oryzae*. It is often isolated from agricultural soils [52] and can ferment various plant materials, including grain crops’ phytomass. Thus its main contribution to the notill soil mycobiome could be due to this ability. As for the *Talaromyces* genus, i.e., another dominant fungal genus with differentially increased abundance in the notill soil, it occurs in various environments, e.g., soil, air, living or rotten plants, and indoors, some of them causing mycoses [53,54]. It is a very species-rich genus, with new species being regularly discovered [55]; despite its diverse physiology and environmental preferences, it is safe to assume that in our soils its representatives (*T. rugulosus, T. albobiverticillius, T. thermophilus, T. luteus, T. ohiensis,* and some unclassified *Talaromyces*) were mostly involved also in straw residue decomposition [56].

Our finding that the 5–15 cm cropped soil mycobiome showed much fewer tillage-related differential abundance, i.e., the increased relative abundance of only two genera (*Exophiala* and *Neonectria*) in the notillage soil, apparently suggests the closer association of the genera with wheat root litter.

### 4.6. The Mycobiome α- and β-Biodiversity

The Shannon α-biodiversity index (calculated on the fungal OTUs basis, with a mean of 4.4) reported here is very close to the ones obtained in some other studies [57], where the reduced tillage displayed no effect on this index. Recently it was reported that in the globally longest experiment, the undisturbed grassland soil, as compared with the notill one, was associated with the higher Shannon diversity of fungal phylotypes [58]; whereas in our study this index, albeit slightly higher, did not differ significantly from the no-till soil due to the considerable variation between soil replicates, which was also observed by other researchers [57].

Other researchers in a three-year-long experiment in China estimated both Shannon and Simpson indices of soil fungal community to be higher under the no tillage as compared with the conventional one [20]. Notably, even within one study, fungal richness in two of the locations could be higher in the no till soil as compared with the conventionally tilled one, being unaffected in the third location [21]. Our finding that the indices for the 0–5 cm mycobiome were the same as in the respective layer of the undisturbed soil, strongly suggests that for some environmental contexts the indices cannot adequately indicate the effects: in our study, rather a big number of fungal taxa (at different taxonomical levels) were found to be differentially abundant due to soil tillage, thus clearly indicating shifts in the mycobiome composition and structure. Our results are in line with the conclusion that estimates of biodiversity do not capture important facets of community adaptation to land management change adequately, which was reached by other researchers before [57] in an experimental study, and in a recently published meta-analysis (conducted on 43 peer-reviewed articles from around the world) to examine the effects of no tillage on soil microbial diversity [59], which concluded that no tillage caused no significant change to fungal diversity.

Yet the soil mycobiome OTUs’ richness was reported to be decreased under reduced or no tillage [56,60], and in our study, there was also a decrease in actual and potential (Chao-1) richness in both cropped fields; however, other studies reported that richness was weakly correlated to tillage [61]. Our finding that in the 0–5 cm layer the observed and potential fungal OTUs’ richness was markedly (15–18%) lower under conventional tillage as compared with no tillage could have resulted in part from the inversion of soil layers by ploughing and mixing by disking; the effect of seasonality may have also contributed to the pattern. In general, the lower species richness in the conventionally ploughed soil results from the fact that such soil provides a more limited range of niche space for fungi due to poor diversity of plant species, reduced organic matter inputs, and poorer soil aggregation, etc.

Our finding that among all studied soil samples the lowest species richness was found in the 5–15 cm layer of the undisturbed soil illustrates just that, i.e., no soil disturbance, except for penetrating roots and precipitation/soil solution leaching. Moreover, most of the α-biodiversity indices for the undisturbed soil mycobiome, differed between 0–5 and 5–15 cm layers, thus confirming the absence of disturbance by tillage.

As for the absence of mycobiome β-diversity between the CN and NT soils, apparently, 12 years of no tillage so far had not resulted in any difference in contrast to the results of some long-term (50 years) field trials, albeit cropped for corn and soybean [62].

### 4.7. Fungal Taxa Abundance and Soil Properties

Although some dominant phyla were found to show certain association with the basic soil properties (as visualized by the principal components analysis), at the OTU level most of the dominants had a mostly negative correlation, producing the impression that their abundance within the range of the studied soil samples and the corresponding range of soil properties is mainly driven by some other soil properties, like specific chemical nature of plant residues, interactions with other microbiota, etc.

### 4.8. General Comments

Our study showed that no tillage shifted soil mycobiome composition (species richness) closer to the one in the undisturbed soil under natural vegetation, most likely by altering physicochemical properties of the soil environment [63], thus expanding the opportunity space [57] for fungi surviving and thriving. As, by general consensus, species-richer fungal communities or naturally developed communities are regarded as “healthier” [64] in comparison with agriculturally used ones, our results support the notion that switching the long-term conventional tillage fields to the no-till ones can improve soil health [65] and ecosystem productivity.

Notably, the soil mycobiome α-biodiversity indices in our study, except for species richness, did not show any tillage-related differences. However, β-biodiversity, as displayed by the dissimilarity index based on Bray-Curtis distance, showed that both cropped fields were grouped together, being rather far from the undisturbed soil. This result suggests that soil mycobiome in the notill soil was much closer to that in the conventionally ploughed soil, despite the difference in soil organic matter content (Table 1) and wheat phytomass residue input. The finding was rather unexpected, as we anticipated some clear distancing between the cropped soils after more than a decade long different tillage; however, such a situation may have been due to the fact that wheat residue decomposition had progressed already for a month and a half prior to the soil sampling for this study, which made the mycobiomes’ diversity more similar.

The possibility to analyze the undisturbed soil under natural vegetation in the area adjacent to the experimental cropped fields is not often available, or the importance of examining such soils can be overlooked by researchers. We believe that including the undisturbed soil in comparative soil metagenomic studies provides a very important reference, crucial for restoring and sustaining soil microbial biodiversity in the future. Therefore, from the ecological point of view, the inclusion of the undisturbed adjacent soil is a strong positive feature of our study.

As for revealing microbial biomarkers, specific for each tillage treatment, we should reiterate that soil samples for the microbiome analysis were taken once, late in the growing season for the soil under undisturbed vegetation and rather long (*ca.* 1.5 months) after the harvest in the wheat-cropped notill and conventionally tilled fields. Therefore, as we did not examine any temporal dynamics, the taxa we found might be regarded as such only for the same time frame. Adopting the recently proposed ecospace framework with its spatiotemporal continuity [66], ideally, researchers should include temporal dynamics for a formal and structured quantification of environmental variation [67]. Thus the absence of temporal dynamics can be considered somewhat of a drawback in our study. To be fair, though, we should note that a) we managed to find only a few studies about the effect of notill on soil microbiota/microbiome which included seasonal dynamics [45]. Moreover, any attempt to grasp temporal continuity, immediately poses a question of how frequent sampling events should be in order to adequately account for such continuity?

At the same time, we want to emphasise again that, with the aim to observe the tillage treatment effect more clearly, soil sampling for this study was performed 1.5 months after wheat harvesting, to decrease the influence of living plants on soil microbial assemblages and to allow a relatively long period of plant residue decomposition by fungi to reveal the main participants in the process. This can be considered a positive aspect of the study with consequences in agrotechnology improvement.

Here we presented results mainly for the dominant taxa, despite the vast sets of identified OTUs. By doing this we are not sending a message that rare species are ecologically and agriculturally unimportant; far from it, we believe that minor or rare species may contribute to shaping inter-species relations and fine-tuning microbiota adjustment to environmental changes. However, often there is little ecological knowledge that can be linked even to the dominant OTUs, let alone the minor or rare ones [68], especially for fungi, as there is a lack of reference fungal genomes for such complex ecosystems as soils [69]. Consequently, ecological interpretation of OTU/species assemblages assessed by analysing environmental DNA is often speculative, and much effort is yet needed to improve the ecological annotation in the reference databases. Yet, we tend to believe that rare sequences are unlikely to crucially affect ecological context [70]; therefore, irrespective of the total mycobiome richness and land management, the dominants were few in all three soils, summarily accounting for 23% of the sequence reads abundance in the undisturbed soil and for 40% in the ploughed, and 45% in the notill soil mycobiomes, respectively, the main fungal actors in soil functioning at the end of the growing season should be sought among the dominating assemblages.

## 5. Conclusions

The presented results provide the first inventory of the mycobiome diversity in the most common soil type (Chernozem) in the important agricultural area in West Siberia, as specifically shaped by vegetation and soil tillage. We revealed a distinct influence of different tillage types upon the mycobiome assembly. The soil mycobiome under natural steppe vegetation was species richest, with no till following, and ploughed soil displaying the lowest richness, most likely due to the reduced opportunity space for microorganisms. After 12 years of continuous notill management the soil harboured, albeit altered mycobiome, yet still rather close to the one in conventionally tilled soil.

## Figures and Tables

**Figure 1 life-12-01169-f001:**
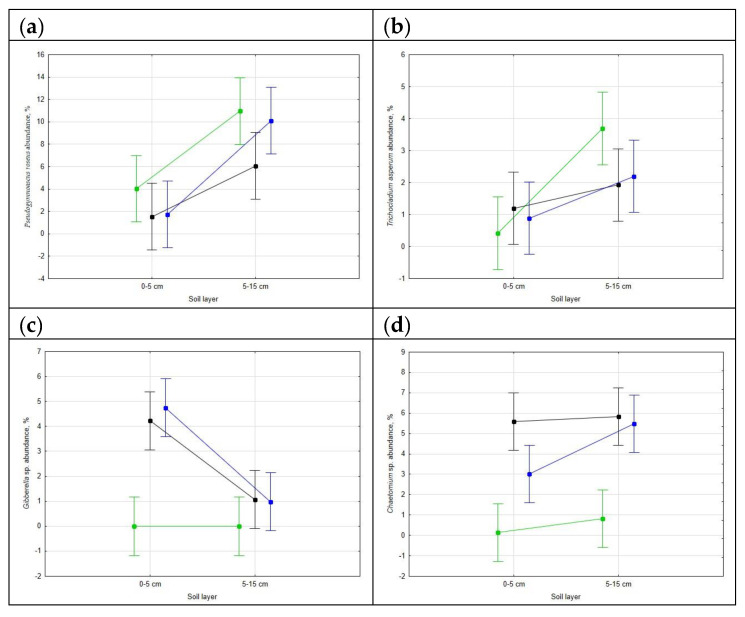
Relative abundance of some dominant fungal OTUs in the undisturbed (shown in green), ploughed (black) and notill (blue) soil (Chernozem) in the forest-steppe zone in the south of West Siberia: *Pseudogymnoascus roseus* (**a**), *Trichocladium asperum* (**b**), *Gibberella* sp. (**c**), *Chaetomium* sp. (**d**), *Mortierella elongata* (**e**) and *Clonostachys rosea* (**f**). The whiskers show 0.95 confidence intervals.

**Figure 2 life-12-01169-f002:**
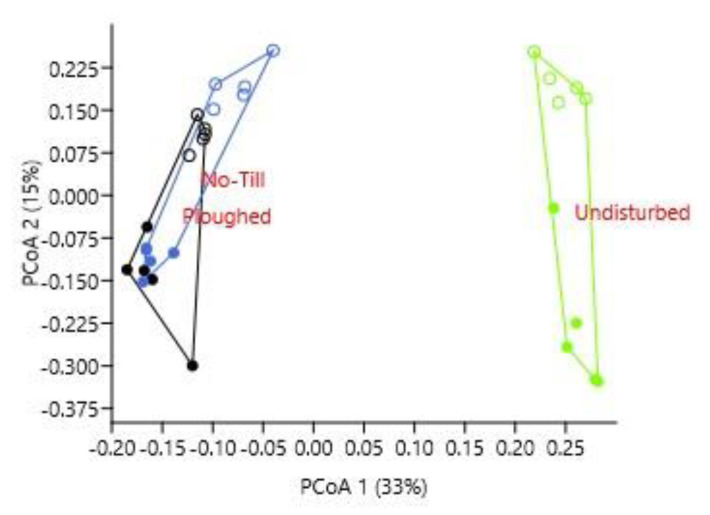
Principal coordinates analysis of the soil mycobiome composition (OTU level, Bray-Curtis distance) under different soil tillage in the forest-steppe zone in West Siberia: location of samples in the plane of the first two coordinates. Symbols used: solid circles indicate samples from the 0–5 cm layer, and open circles indicate samples from the 5–15 cm layer.

**Figure 3 life-12-01169-f003:**
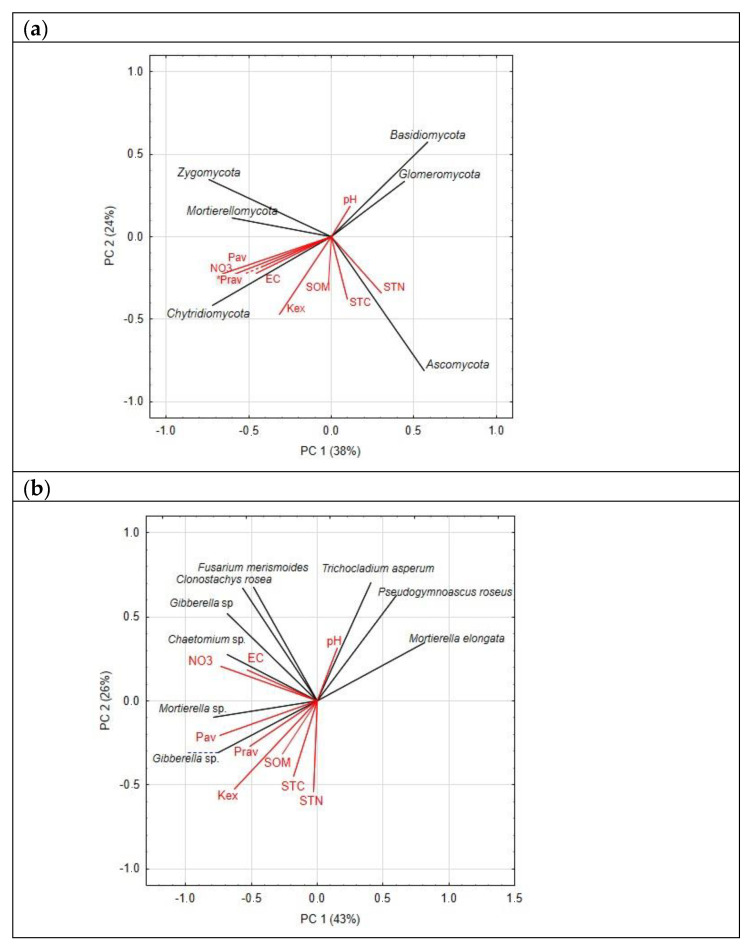
Principal components analysis: location of the dominant phyla (**a**) and OTUs (**b**) in the plane of the first two principal components—factor loadings for the relative abundance of taxa (active variables) and soil properties (supplementary variables); and location of the soil samples (**c**). Abbreviations used: STC—soil total C, STN—soil total nitrogen, SOM—soil organic matter, EC—electric conductivity, Pav—available P_2_O_5_, Prav—readily available *p*, Kex—exchangeable K. Symbols used: solid circles indicate samples from the 0–5 cm layer, and open circles indicate samples from the 5–15 cm layer.

**Table 1 life-12-01169-t001:** Soil properties in different experimental fields in the south of West Siberia.

Property	Undisturbed		Ploughed		No Till	
	0–5 cm	5–15 cm	0–5 cm	5–15 cm	0–5 cm	5–15 cm
Bulk density, g cm^−3^ soil	1.11	1.23	1.02	1.09	1.13	1.19
Soil texture, %						
Sand (2–0.063 mm)	34 ^b #^	34 ^b^	16 ^a^	15 ^a^	15 ^a^	16 ^a^
Silt (0.063–0.002 mm)	41 ^a^	43 ^a^	53 ^b^	51 ^b^	52 ^b^	50 ^b^
Clay (<0.002 mm)	25 ^b^	23 ^a^	31 ^bc^	34 ^c^	33 ^c^	34 ^c^
pH	6.61 ^b^	6.65 ^bc^	6.67 ^bc^	6.82 ^c^	6.29 ^a^	6.75 ^bc^
EC *, µS	239 ^ab^	209 ^a^	286 ^bc^	288 ^bc^	286 ^bc^	330 ^c^
STC, %	4.2 ^b^	3.6 ^a^	4.0 ^bc^	3.8 ^ac^	4.1 ^bc^	3.8 ^ac^
STN, %	0.37 ^c^	0.31 ^a^	0.35 ^c^	0.33 ^bc^	0.33 ^b^	0.29 ^a^
SOM, %	9.7 ^c^	7.8 ^a^	8.9 ^b^	8.4 ^ab^	9.6 ^c^	8.9 ^b^
NO_3_^−^, mg N kg^−1^ soil	2.0 ^a^	1.3 ^a^	3.6 ^b^	3.2 ^b^	4.6 ^c^	4.8 ^c^
Pav, mg P_2_O_5_ kg^−1^ soil	15.2 ^b^	8.6 ^a^	81.5 ^d^	37.7 ^c^	77.4 ^d^	18.4 ^bc^
Prav, mg P_2_O_5_ kg^−1^ soil	0.55 ^ab^	0.22 ^a^	1.74 ^bc^	0.39 ^a^	2.74 ^c^	0.15 ^a^
Kex, mg K_2_O kg^−1^ soil	577 ^d^	283 ^a^	781 ^e^	494 ^c^	726 ^e^	354 ^ab^

^#^ Different letters in rows indicate statistically significant difference at *p* ≤ 0.05 (Fisher’s LSD test). * Abbreviations used: STC—soil total C, STN—soil total nitrogen, SOM—soil organic matter, EC—electric conductivity, C/N—the ratio of C and N in soil, Pav—available *p*, Prav—readily available *p*, Kex—exchangeable K.

**Table 2 life-12-01169-t002:** Relative abundance (%, mean) of the dominant fungal phyla and classes in Chernozem 0–5 and 5–15 cm layers in the experimental fields in the south of West Siberia.

Taxon	Undisturbed		Ploughed		No Till	
	0–5 cm	5–15 cm	0–5 cm	5–15 cm	0–5 cm	5–15 cm
**Phylum level**						
*Ascomycota*	85.8 ^b,1^	82.1 ^ab^	82.9 ^ab^	78.6 ^a^	80.4 ^a^	80.0 ^a^
*Basidiomycota*	7.9 ^c^	6.5 ^bc^	4.3 ^ab^	5.4 ^abc^	3.1 ^a^	4.0 ^ab^
*Zygomycota*	2.4 ^a^	5.0 ^ab^	6.3 ^b^	6.8 ^b^	10.7 ^c^	7.6 ^bc^
*Mortierellomycota*	1.1 ^a^	3.4 ^b^	4.1 ^bc^	6.3 ^c^	3.4 ^b^	5.3 ^bc^
*Chytridiomycota*	0.4 ^a^	0.4 ^a^	1.1 ^c^	0.7 ^b^	1.1 ^c^	0.7 ^b^
*Glomeromycota*	0.8 ^b^	1.0 ^b^	0.0 ^a^	0.3 ^a^	0.1 ^a^	0.9 ^b^
*un.* ^3^ *Fungi*	0.9 ^ab^	1.2 ^b^	0.9 ^ab^	1.2 ^b^	0.7 ^a^	0.9 ^ab^
**Class level**						
*Sordariomycetes*	15.6 ^a^	15.2 ^a^	51.2 ^b^	46.0 ^b^	43.6 ^b^	41.4 ^b^
*Dothideomycetes*	30.2 ^b^	18.3 ^a^	18.7 ^a^	15.3 ^a^	16.8 ^a^	17.1 ^a^
*Eurotiomycetes*	19.1 ^c^	25.7 ^d^	5.3 ^a^	11.4 ^b^	11.1 ^b^	14.4 ^bc^
*Agaricomycetes*	4.6 ^b^	5.2 ^b^	3.3 ^ab^	3.9 ^b^	1.3 ^a^	2.3 ^ab^
*Leotiomycetes*	5.6	8.1	5.6	4.4	6.7	5.8
*Mucoromycotina*_is ^2^	2.4 ^a^	4.6 ^a^	6.2 ^b^	6.2 ^bc^	10.6 ^c^	7.3 ^bc^
*Mortierellomycetes*	1.1 ^a^	3.4 ^b^	4.1 ^b^	6.3 ^c^	3.4 ^b^	5.3 ^bc^
*unc_Ascomycota*	6.1 ^bc^	10.4 ^c^	1.0 ^b^	0.5 ^a^	1.1 ^ab^	0.5 ^a^
*Tremellomycetes*	1.7 ^b^	0.4 ^a^	0.4 ^a^	0.9 ^ab^	1.0 ^ab^	1.1 ^ab^
*Orbiliomycetes*	2.2 ^b^	2.6 ^b^	0.1 ^a^	0.1 ^a^	0.1 ^a^	0.1 ^a^
*Pezizomycotina*_is	3.4 ^b^	0.1 ^a^	0.1 ^a^	0.1 ^a^	0.3 ^a^	0.1 ^a^
*Glomeromycetes*	0.8 ^bc^	1.0 ^c^	0.0 ^a^	0.3 ^ab^	0.1 ^a^	0.9 ^bc^

^1^ Different letters after the values in rows indicate that the values differ at *p* ≤ 0.05 level (Fisher’s LSD test); the absence of letters in a row indicates that there was no difference. ^2^ “is” stands for insertae sedis. ^3^ un. stands for unclassified.

**Table 3 life-12-01169-t003:** Relative abundance (%, mean) of the dominant fungal genera in Chernozem 0–5 and 5–15 cm layers) of the experimental fields in the south of West Siberia.

Genus	Undisturbed		Ploughed		No Till	
	0–5 cm	5–15 cm	0–5 cm	5–15 cm	0–5 cm	5–15 cm
*Chaetomium*	0.2 ^a,1^	0.9 ^a^	8.2 ^d^	6.8 ^cd^	4.3 ^b^	5.9 ^c^
*Clonostachys*	0.3 ^a^	0.1 ^a^	1.1 ^b^	2.7 ^c^	2.3 ^c^	2.7 ^c^
*Cordyceps*	0.0 ^a^	0.2 ^ab^	0.7 ^ab^	2.0 ^c^	1.0 ^b^	2.8 ^c^
*Devriesia*	5.6 ^b^	0.2 ^a^	0.0 ^a^	0.0 ^a^	0.0 ^a^	0.0 ^a^
*Exophiala*	1.0 ^a^	0.7 ^a^	0.7 ^a^	0.6 ^a^	2.5 ^c^	1.6 ^b^
*Fusarium*	0.5 ^a^	1.5 ^ab^	2.6 ^b^	5.2 ^c^	2.4 ^ab^	4.7 ^c^
*Gibberella*	1.7 ^a^	0.7 ^a^	7.3 ^c^	5.4 ^bc^	11.2 ^d^	5.3 ^bc^
*Herpotrichia*	1.5 ^a^	0.0 ^a^	0.8 ^a^	0.0 ^a^	3.1 ^b^	0.7 ^a^
*Humicola*	0.6 ^a^	3.9 ^c^	1.2 ^a^	2.0 ^ab^	0.9 ^a^	2.2 ^b^
*Hypocrea*	0.3 ^a^	0.3 ^a^	1.9 ^b^	2.2 ^b^	2.3 ^b^	2.1 ^b^
*Knufia*	4.4 ^b^	0.1 ^a^	0.0 ^a^	0.0 ^a^	0.2 ^a^	0.0 ^a^
*Lecythophora*	0.4 ^a^	0.0 ^a^	2.2 ^c^	0.6 ^a^	1.4 ^b^	0.4 ^a^
*Lophiostoma*	0.1 ^a^	0.1 ^a^	0.6 ^a^	0.1 ^a^	2.5 ^b^	0.2 ^a^
*Metarhizium*	0.4 ^a^	0.1 ^a^	0.8 ^ab^	1.7 ^c^	1.3 ^bc^	1.3 ^bc^
*Mortierella*	3.4 ^a^	8.0 ^b^	8.2 ^bc^	11.2 ^c^	10.5 ^bc^	10.7 ^c^
*Neonectria*	0.0 ^a^	0.1 ^a^	0.4 ^a^	0.9 ^b^	0.9 ^b^	1.9 ^c^
*Penicillium*	6.7 ^c^	5.3 ^abc^	1.9 ^a^	5.7 ^bc^	2.5 ^ab^	3.4 ^abc^
*Phialocephala*	0.9 ^a^	0.6 ^a^	2.1 ^b^	1.3 ^ab^	1.5 ^ab^	1.4 ^ab^
*Podospora*	1.5 ^ab^	0.8 ^a^	11.0 ^d^	4.3 ^c^	3.7 ^bc^	1.5 ^ab^
*Preussia*	0.5 ^ab^	0.1 ^a^	2.0 ^c^	1.4 ^b^	1.1 ^b^	0.8 ^ab^
*Pseudogymnoascus*	4.6 ^ab^	11.0 ^c^	2.6 ^a^	7.2 ^bc^	2.3 ^a^	10.4 ^c^
*Rhinocladiella*	3.3 ^b^	5.1 ^c^	0.0 ^a^	0.0 ^a^	0.0 ^a^	0.0 ^a^
*Rhizopus*	0.0 ^a^	0.0 ^a^	1.2 ^ab^	0.8 ^ab^	3.1 ^c^	1.5 ^b^
*Talaromyces*	0.3 ^a^	0.0 ^a^	1.5 ^ab^	1.6 ^ab^	4.0 ^c^	3.1 ^bc^
*Tetracladium*	0.0 ^a^	0.8 ^ab^	1.6 ^abc^	1.0 ^abc^	2.3 ^bc^	2.8 ^c^

^1^ Different letters in rows indicate that the values are different (*p* ≤ 0.05, Fisher’s LSD test); the absence of letters after the values in a row indicate that there was no difference.

**Table 4 life-12-01169-t004:** Alpha-biodiversity indices (calculated on the OTU’s basis) of mycobiomes in the Chernozem (0–20 cm) of the experimental fields in the south of West Siberia.

Index	Undisturbed		Ploughed		No till	
	0–5 cm	5–15 cm	0–5 cm	5–15 cm	0–5 cm	5–15 cm
OTU richness	710 ^c, 1^	424 ^a^	497 ^a^	565 ^b^	584 ^b^	539 ^b^
Chao-1	753 ^c^	433 ^a^	545 ^ab^	602 ^b^	629 ^b^	569 ^b^
Simpson (S)	0.98	0.91	0.97	0.98	0.98	0.91
Shannon’s	4.7 ^b^	3.8 ^a^	4.4 ^b^	4.6 ^b^	4.6 ^b^	4.2 ^ab^
Evenness	0.16 ^ab^	0.12 ^a^	0.18 ^b^	0.19 ^b^	0.17 ^ab^	0.15 ^ab^
Equitability	0.72 ^b^	0.63 ^a^	0.72 ^b^	0.73 ^b^	0.72 ^b^	0.66 ^ab^
Berger-Parker	0.09 ^a^	0.23 ^b^	0.08 ^a^	0.07 ^a^	0.07 ^a^	0.20 ^ab^
Dominance (1-S)	0.02	0.09	0.03	0.02	0.02	0.09

^1^ Different letters in rows indicate that the values are different (*p* ≤ 0.05, Fisher’s LSD test); the absence of letters after the values in a row indicate that there was no difference.

## Data Availability

The read data reported in this study were submitted to the GenBank under the study accession PRJNA845814.

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
