# Peer review of "Soil Mycobiome Diversity under Different Tillage Practices in the South of West Siberia"

_life, 2022, doi:10.3390/life12081169_

Round 1

Reviewer 1 Report

Please made corrections as mentioned in the original manuscript.

Author Response

Point 1

Please provide full crop history since 2009.

Response 1

We inserted the following text: “were rotationally cropped (three years for spring wheat followed by a year of spring rapeseed) and”…

Point 2

please super script: -1

Response 2

The script corrected.

Thank you very much for your reviewing of the manuscript and its approval for the publication!

Reviewer 2 Report

As general comment the work is well written and designed with relevant results.

In general terms the topic of the article is interesting, the methodology is explicitly presented and the results reported are interesting.

The structure of the paper is correct.

In my opinion, the abstract is too general, please reframe.

The introduction chapter should end with a paragraph indicating the purposefulness of the conducted research. Authors should clearly define the purpose of the work and formulate research hypotheses.

Materials and method section is well described and correspond to the aim set out in the manuscript. The tables and figures clearly presenting the obtained results with their appropriate interpretation.

The statistical calculation methods used in the research make the obtained results reliable and provide a basis for drawing correct conclusions.

The references are sufficient and necessary.

The paper needs some editorial corrections.

I recommend the publication of this manuscript in the Life journal after minor revisions.

Author Response

Point 1

In my opinion, the abstract is too general, please reframe.

Response 1

We agree, and to this end we somewhat rephrased the middle part of the abstract, (trying to comply, anyway, with the 200-words limit) by inserting the following:

“Several dominant genera (Mortierella, Chaetomium, Clonostachys, Gibberella, Fusarium and Hypocrea) had increased abundance in both cropped soils as compared with the undisturbed one, and therefore can be safely assumed to be associated with wheat residues.”

… and deleting “Overall, 137 genera had differential abundance in the studied soils.”.

Point 2

The introduction chapter should end with a paragraph indicating the purposefulness of the conducted research. Authors should clearly define the purpose of the work and formulate research hypotheses.

Response 2

The version of the manuscript you reviewed ends in the following paragraph: «The aim of the study was to reveal mycobiome composition and structure in Chernozem under natural vegetation or cropped for wheat by conventional or no tillage in the long-term field experiment in the Novosibirsk region, Russia, by using ITS region DNA diversity”. We are afraid we cannot add more purposefulness to this statement, can we?

As for the research hypothesis, frankly speaking, prior to the start of the study we did not fathom something more specific than some differences in the soil mycobiome of the fields to study. The main aim was to inventory and to see where we stand with the results and where to go further (currently, for instance, we are trying to analyze comprehensively some actinobacteria strains from the no-till soil). To come up with some hypothesis now does not seem appropriate, sorry for this.

Thank you very much for your reviewing of the manuscript and for your suggestions to improve it!

This manuscript is a resubmission of an earlier submission. The following is a list of the peer review reports and author responses from that submission.